# Polynomial Regression as a Task for Understanding In-context Learning Through Finetuning and Alignment

**Max Wilcoxson** [* 1]  **Morten Svendgård** [* 1]  **Ria Doshi** [* 1]  **Dylan Davis** [* 1]  **Reya Vir** [1]  **Anant Sahai** [1]

## Abstract

Simple function classes have emerged as toy problems to better understand in-context-learning in transformer-based architectures used for large language models. But previously proposed simple function classes like linear regression or multi-layer-perceptrons lack the structure required to explore things like prompting and alignment within models capable of in-context-learning. We propose univariate polynomial regression as a function class that is just rich enough to study prompting and alignment, while allowing us to visualize and understand what is going on clearly. Code can be found at https://github.com/MSNetrom/in-context-poly-playground.

## 1. Introduction and Motivation

Once Brown et al. (2020) called out that that pretrained large language models (LLMs) have the emergent ability to do in-context learning, this opened the door to using prompting to direct LLM behavior. Ouyang et al. (2022) showed how appropriate fine-tuning could strengthen prompting into instruction-following and also emphasized the need for alignment. Although this line of work started in the very rich domain of natural language using very large models, the computational challenges of training such large models from scratch were soon recognized. Garg et al. (2023) zoomed in on the core issue of in-context learning and proposed using very simple function classes (where we can often hand-craft optimal approaches to compare against) to study things without having to use very large models or tokenization. However, prior examples of simple function classes like linear regression, sparse linear regression, de-

cision trees, and multi-layer-perceptrons[1] (MLPs) are too unstructured to be able to naturally capture ideas related to alignment or jailbreaking, or to systematically explore fine-tuning. To remedy this, we introduce a toy based on univariate polynomial regression — this is like linear regression, but we implicitly lift via the Chebyshev polynomials.

Univariate polynomials are easily visualizable, but for a toy problem to be useful as an object of study, it should exhibit at least some of the interesting behaviors of reality. Specifically we look at the following questions:

- **Can polynomials be learned in-context?** Yes.

- **How does LoRA perform compared to soft prompting for such functions? Does this match what we see in LLMs?** Just as for LLMs, **LoRA** is better with a comparable amount of parameters.

- **Can we capture the idea of alignment, refusal, and jailbreaking using univariate functions?** Yes!

- **How does adding jailbroken examples to the context window affect model alignment?** Increasing the number of jailbroken examples leads to worse alignment, matching the behavior of LLMs.

### 1.1. Parameter efficient fine tuning

**Soft prompting** is the concept of adding learnable task-specific embeddings to the start of the embedded input-sequence (Lester et al., 2021). Because the complement of the input data's embedding image is significant, the added flexibility can give an advantage over normal hard prompting. Another method is **LoRA** (Low-Rank Adaptation of Large Language Models) which "freezes the pre-trained model weights and injects trainable low-rank decomposition matrices into each layer of the Transformer architecture" (Hu et al., 2021). For $h = W_0 x$, the modified pass yields:

$$h = W_0 x + \Delta W x = W_0 x + BAx$$

---

[*]Equal contribution , arbitrary listing order. [1]EECS, University of California, Berkeley, USA. Correspondence to: Anant Sahai <sahai@eecs.berkeley.edu>.

*Proceedings of the $1^{st}$ Workshop on In-Context Learning at the $41^{st}$ International Conference on Machine Learning*, Vienna, Austria. 2024. Copyright 2024 by the author(s).

---

[1]Interestingly, MLPs *are* a very natural way to study chain-of-thought (CoT) prompting (Li et al., 2024). But this actually helps us understand how CoT is different from "instruction-following" style prompting.

Where $B \in \mathbb{R}^{d \times r}$, $A \in \mathbb{R}^{r \times k}$, and the rank $r \leq \min(d, k)$. We apply **LoRA** to the attention matrices.

## 1.2. Alignment

Within language contexts, there are many different motivations for wanting to align a model's outputs. In LLMs, alignment can be understood as the process of fine-tuning to have the model produce more desirable outputs. When the alignment goal is understood in terms of refusing to engage in certain behavior, the concept of jailbreaking captures when that refusal goal is foiled. We study how a model pretrained to in-context learn a functional class can have its behavior aligned to a new but related task that models refusal. The goal is to both showcase that a model could be aligned within this toy domain as well as make the case that studying alignment in this context is reasonable and can give us a better understanding of alignment.

## 2. Related Works

**Prompting:** There is a large literature of relevant work regarding phenomena in PEFT. In Xu et al. (2023) **LoRA** is seen to perform better than prefix-tuning (and **soft prompting**) on all GLUE metrics. We also observe that **LoRA** performs better than **soft prompting** on our polynomial tasks. Petrov et al. (2024) looks at the toy problem of sorting numbers in increasing order, then finetuning using prefix-tuning for learning to sort in descending order. Prefix-tuning struggles to complete this task, which shows the power of toy problems for more crisply highlighting the limitations of finetuning methods, and our work fits in this broad vein.

**Alignment:** While there has been a lot of research done in alignment for LLMs, the capabilities of transformers in learning function classes when constraints are introduced remains unclear. Scaling laws for LLMs have been studied and have revealed that models show a clear power-law scaling behavior with respect to context lengths (Xiong et al., 2023). ICL performance improves with increased number of demonstrations as well as increased length of instructions (Li et al., 2023).

## 3. Methodology

### 3.1. Models and Training Configurations

Following Garg et al. (2023), the model used is a GPT2-style model, with 6 layers, 4 heads, and an embedding dimension of 128. The model has a total of about 1.2 million parameters. See appendix A.3 for details.

### 3.2. Univariate Chebyshev Polynomials

The constructed tasks follow the in-context learning structure as in Garg et al. (2023), which is of the form:

$$(x_1, y_1, x_2, y_2, ..., x_{query})$$

where $x$-values are scalars sampled from $\mathcal{U}(-1, 1)$ and the $y$-values are also scalars.

Random linear combinations of Chebyshev polynomials provide an effective way to sample well-behaved polynomials that do not have extreme y-values. Some properties of Chebyshev polynomials are:

- For $x \in [-1, 1]$ we have $y \in [-1, 1]$.

- All roots are on the interval $x_{root} \in [-1, 1]$

### 3.3. Chebyshev Linear Combinations

The base-models are trained on a linear combinations of our Chebyshev polynomials. Chebychev polynomials of the first kind can be recursively defined by:

$$T_0(x) = 1, \ \ T_1(x) = x$$
$$T_{n+1}(x) = 2x \, T_n(x) - T_{n-1}(x).$$

We sample weights from a standard normal distribution, which gives:

$$p(x) = \sum_{i=0}^{b} c_i T_i(x), \ c_i \sim \mathcal{N}(0, 1) \tag{1}$$

For pretraining, we sample a maximum degree $b$ uniformly from $[0, 11]$ and create a normal random linear combination of Chebyshev polynomials of degree at most $b$. The value of $y$ is approximately marginally independent of $x$, as sampling many points from many different polynomials generated from this method leads to a distribution that does not vary much with $x$, as shown in Figure $1a$.

### 3.4. Specific Degree and Fixed Coefficients

For parameter-efficient finetuning, we only sample weighted combinations of Chebyshev polynomials of degrees at most 5. We choose 5 since it is in the middle of our pretraining degree range of 0 to 11. We also fix between 0 to 5 of the coefficients $c_i$ to be 1. This leads to a marginal distribution that does vary with $x$ (see Figure 1b). This creates a task which occupies a subspace of the original training distribution, providing more clear room for improvement.

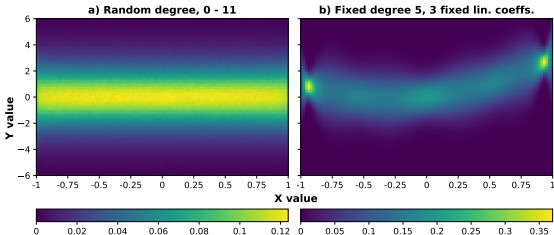

Figure 1: Sampled density heatmap of joint (x, y) distribution for uniformly random x-values and linear combinations of Chebyshev polynomials. See appendix for a more detailed explanation (section A.1). (a) Polynomials of random degree up to 11. The distribution of $y$ is approximately independent of $x$. (b) Polynomials of degree 5 (**specifically**), but the first 3 linear coefficients are fixed. The distribution of $y$ is not independent of $x$.

### 3.5. Refusal as a Toy Model of Alignment

For testing refusal-style alignment, a task based on the linear combination of Chebyshev polynomials (see section 3.3) is edited to have the $y$-values clamped at a certain threshold. Instead of predicting the polynomial $p(x)$, the model should now predict $min(p(x), T)$.

This task was chosen at it allows our model to leverage what it already knows while also learning a new behavior, and it resembles many real world alignment objectives such as answering questions so long as the desired result does not exceed some level of toxicity or offensiveness.

## 4. Results

### 4.1. Learning Polynomials In-context

As seen in Figure 2, it is possible for a GPT2-style model as described in section 3.1, to learn to do regression for linear combinations of Chebyshev polynomials. As the context increases, the predictions get better. We use linear regression in the basis formed by Chebyshev polynomials of up to degree 11, with and without ridge regularization, as baselines (**polynomial regression** and **ridge-regularized polynomial regression**) (See section A.4). The model performs better than the baselines in low sample regime, indicating better conditioning. In high data regime, the analytical baseline **polynomial regression** is better, as perfectly zero error is difficult to achieve with function approximation.

### 4.2. LoRA vs Soft Prompting

In Figure 3, the performance of **LoRA** and **soft prompting** is compared on the task from section 3.4. As the number of fixed coefficients is increased, the sampled polynomials become less random, and the distribution of $y$ becomes

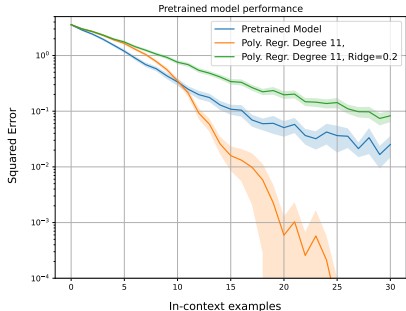

Figure 2: Performance of pretrained model on normal random linear combinations of Chebyshev polynomials of a random degree between 0 and 11. **Polynomial regression** and **ridge-regularized polynomial regression** are used as baselines (See section A.4). Using **polynomial regression**, 12 examples are theoretically needed to achieve $0 \approx 10^{-\infty}$ error for a degree 11 polynomial. **Shaded areas** represent the 95% bootstrap confidence interval (A.5). See Appendix A.9 for plot using linear y-scale.

more dependent on $x$ (see section 3.4). As the distribution becomes less random, **50-pair soft prompting** performs better. Surprisingly, with zero fixed coefficients, **50-pair soft prompting** performs significantly worse than the untrained baseline. Since this task distribution is only slightly more narrow than the mixed degree training distribution, the model may need to learn to ignore the randomly initialized prompts, leading to worse performance. (See Appendix A.8 for a learning rate ablation to isolate effect of LR selection as a possible explanation). **50-pair soft prompting** performs the best with 5 shared coefficients, where there are five fixed points for the **soft prompts** to learn, which supports that **soft prompting** benefits from more predictable distributions (See Appendix A.2).

### 4.3. Alignment Through Eval-Time Context Clamping

The first set of alignment experiments uses the task from section 3.5, and are shown in figure 4. The model has never seen clamped $y$ values during train time, but is able to learn this clamping behavior quite well. However, from our regression baseline, we see that alignment behavior is fairly natural to our task provided we have enough example points. That being said, when our model tries to predict the function in regions that lack in-context examples (such as near the edges), it can revert to unaligned behavior. This shows that the model is in some sense being bounded in-context, but is not truly learning the new aligned behavior. These results coincide with previous work done on larger models and may help explain why many-shot in-context learning works so well (Agarwal et al., 2024).

See appendix A.11 for a comparison between performance

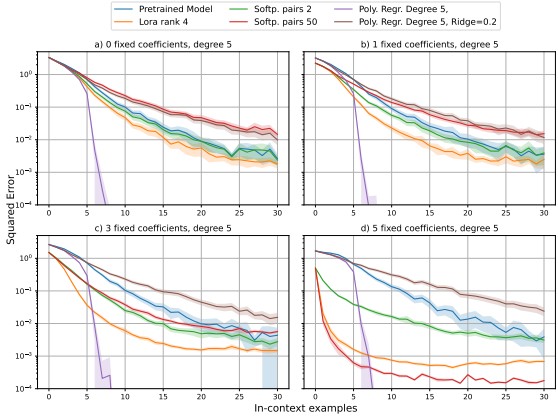

Figure 3: Performance of finetuning methods for normal random linear combinations of Chebyshev polynomials of degree at most 5, and with the first $n$ first linear combination coefficients fixed. As the number of fixed coefficients increases, $y$ becomes more dependent on $x$. **Polynomial regression** and **ridge-regularized polynomial regression** are used as baselines (See section A.4). **Shaded areas** represent the 95% bootstrap confidence interval (A.5). See Appendix A.9 for plot using linear y-scale.

on our toy problem alignment and tasks solved by LLMs, as well as the effects of model size.

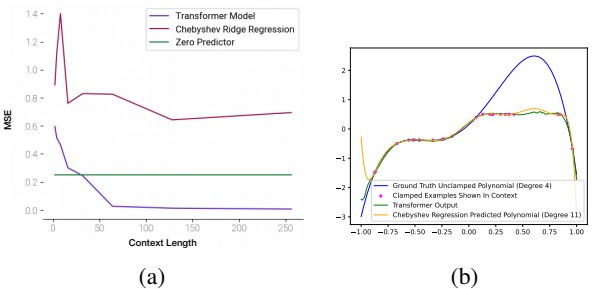

Figure 4: (4a) Increased context length increases performance for clamped polynomials, although the model were not trained to do this. Note that on points above the threshold loss increases, whereas the same is not true for points below the threshold. (4b) Adding in a clamped context leads to good predictions in which our predicted polynomial looks like a smoothed version of the actual clamped polynomial. This example is a degree 4 polynomial with 25 in-context examples of which 9 are clamped.

### 4.4. Alignment Through Finetuning and Jailbreaking

Following our examination of alignment in-context, we finetuned a model to complete our established alignment task as shown in Figure 5a. We then decided to examine the potential for our model to be jailbroken and compare the behavior of our model with the behavior observed in LLMs.

We query our model with a context window of malicious (unclamped) examples, and ask it to provide the $y$ value for the last point, whose ground truth $y$ value lies above the threshold. In Figure 5b, we examine the effect of context length on the model's susceptibility to being jailbroken. As the context length increases, we see the model is more likely to output unaligned responses. These results match those from previous work examining the effect of jailbreaking in LLMs (Wei et al., 2024; Anil et al., 2024).

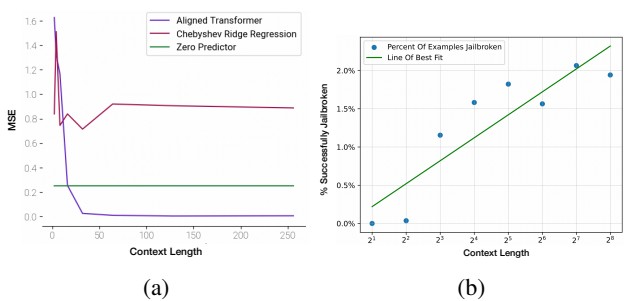

Figure 5: (5a) The transformer model is able to perform the task of clamping values in-context after being additionally finetuned. (5b) As the context length increases, so does the number of jailbroken examples. Similar to language models, as the number of jailbroken examples increases, it becomes more likely for alignment to be broken.

## 5. Limitations

Just as in Garg et al. (2023), a core limitation is establishing relevance between our results and LLMs beyond parallel and similar behavior.

## 6. Conclusion

In this paper, the toy problem of learning linear combinations of Chebyshev polynomials is demonstrated to be useful for exploring LLM-relevant phenomena like alignment and prompting, while using dramatically less compute.

For PEFT, we see that performs better than **soft prompts** in general, as seen in language tasks in LLMs. Additionally, the performance of **soft prompting** seems to vary with the prompt dimension and benefit from a narrower task distribution.

Alignment experiments show that we were able to align our base model in context with a sufficiently large number of aligned examples. We could also jailbreak our aligned model with enough jailbroken examples. These results match those found in LLM domains in (Agarwal et al., 2024) and (Anil et al., 2024). This combined with our other results suggest that the simple task of polynomial regression could be useful gaining understanding of in-context learning in large language models.

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

## A. Appendix

### A.1. Distribution of linear combination of Chebyshev polynomials

In this section, we analyze the distribution of a random polynomial of the following form:

$$p(x) = \sum_{i=a}^{b} c_i T_i(x), \quad c_i \sim \mathcal{N}(0, \sigma^2), \ b \sim \text{Unif}\{a, a+1, \ldots, c\}$$

**Let $f(w)$ be the PDF of $p$**. Then we can write:

$$f(w) = \sum_{b=a}^{c} f(w \mid b) P(b) = \frac{1}{c-a+1} \sum_{b=a}^{c} f(w \mid b)$$

If $b$ is given, $p$ is a sum of normal variables with $\mathbb{E}[p \mid b] = 0$ and $\text{Var}[p \mid b] = \sigma^2 \sum_{i=a}^{b} T_i^2$, and we get:

$$f(w|b) = \frac{1}{\sqrt{2\pi\sigma^2 \sum_{i=a}^{b} T_i^2}} \exp\left(-\frac{w^2}{2\sigma^2 \sum_{i=a}^{b} T_i^2}\right)$$

Combining the two above expressions, we get:

$$f(w) = \frac{1}{c-a+1} \sum_{b=a}^{c} \frac{1}{\sqrt{2\pi\sigma^2 \sum_{i=a}^{b} T_i(x)^2}} \exp\left(-\frac{w^2}{2\sigma^2 \sum_{i=a}^{b} T_i(x)^2}\right)$$

**We can also explicity find the expected value and variance**:

$$\mathbb{E}[p] = \sum_{b=a}^{c} \mathbb{E}[p \mid b] P(b) = 0$$

The law of total variance states:

$$\text{Var}[X] = \mathbb{E}[\text{Var}(X \mid Y)] + \text{Var}[\mathbb{E}(X \mid Y)]$$

By using this, we get:

$$\text{Var}[p] = \mathbb{E}\left[\sigma^2 \sum_{i=a}^{b} T_i^2\right] + \text{Var}[0] = \sigma^2 \sum_{j=a}^{c} \mathbb{E}\left[\sum_{i=a}^{j} T_i^2\right] P(b=j) = \frac{\sigma^2}{c-a+1} \sum_{j=a}^{c} \sum_{i=a}^{j} T_i(x)^2$$

$$\text{Var}[p] = \frac{\sigma^2}{c-a+1} \sum_{i=a}^{c} T_i(x)^2 (c-i+1)$$

The standard deviation and PDF as a function of $x$, is plotted in figure 6.

We would like to thank *Ovidiu-Neculai Avadanei*, PhD student at the *Berkeley Department of Mathematics*, for some inputs in these analysis.

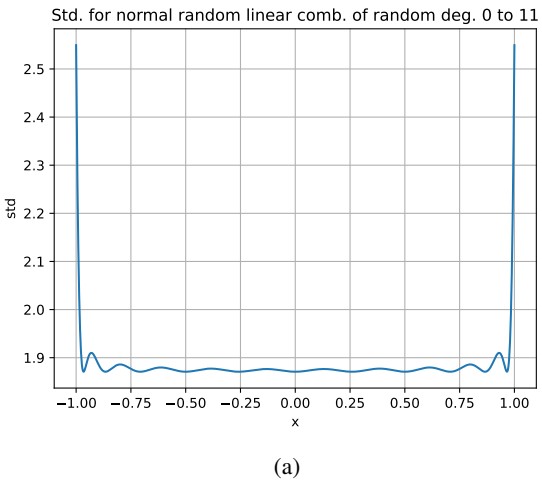
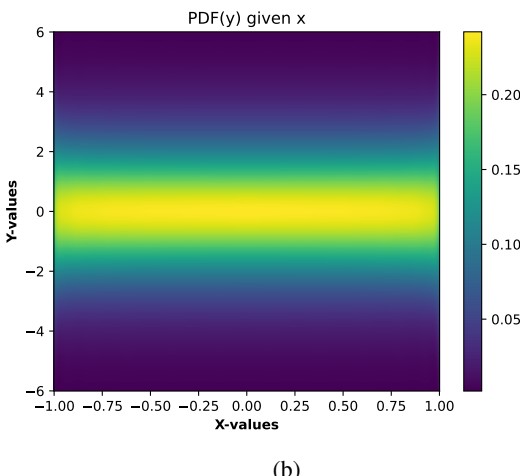

(a)                                        (b)

Figure 6: (6a) The standard deviation as a function of x for normal random linear combination of Chebyshev polynomials between degree 0 and a uniformly random degree from 0 to 11. (6b) The corresponding distribution of y, dependent on the choice of x. Since $x$ is uniformly sampled, this distribution is the same as the sampled joint $x$, y distribution in Figure 1. Although the intensity values are scaled up in this example as the probability needs to "sum" to 1 for every given $x$, instead of for all combinations of $x$, $y$.

### A.2. Shared Points in 5 Shared Coefficients Function Class

When using a linear combination of Chebyshev polynoials of degree 5, with the 5 first linear coefficients fixed, there will be 5 fixed points, independent of the choise of the last coefficient. This is visualized in Figure 7. By fixing the first 5 coefficients to 1 in equation 1 ($c_i = 1$) we get that the difference between any of the sampled functions are $l \cdot T_5(x)$, and therefore 0 in 5 points (as $T_5(x)$ has 5 zeros). This can be shown:

$$p(x) = c_5 T_5(x) + \sum_{i=0}^{4} T_i(x), \quad c_5 \sim \mathcal{N}(0, 1)$$

Let $h(x)$ be any function, then we have:

$$f(x) = c h_k(x) + \sum_{i=a}^{b} h_i(x)$$

For $\hat{f}(x) = \hat{c} h_k(x) + \sum_{i=a}^{b} h_i(x)$ and $f(x) = \hat{f}(x)$:

$$f(x) - \hat{f}(x) = (c - \hat{c}) h_k(x) = 0$$

$$h_k(x) = 0$$

In our case this gives $T_5(x) = 0$, meaning that $p(x)$ has fixed points in the five $x$-values where $T_5(x)$ have roots.

### A.3. Model and training information

This is a supplement to information given in section 3.1.

Overview of model and training configurations:

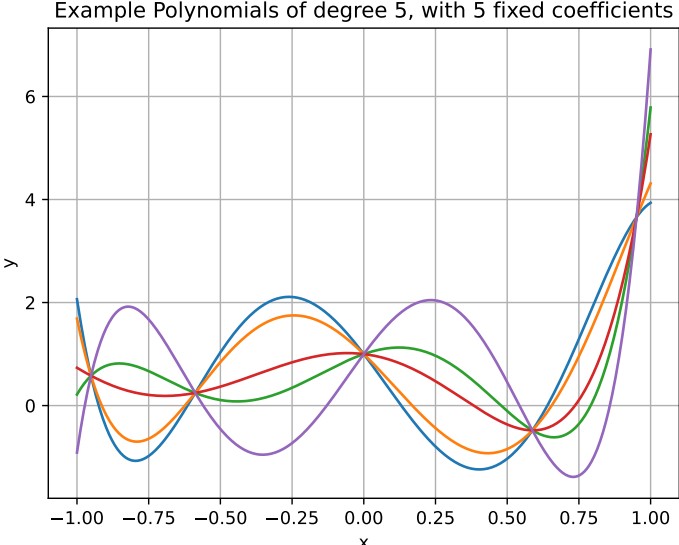

Figure 7: Examples of linear combinations of Chebyshev polynmoials of degree 5, with 5 of 6 linear coefficients fixed.

- For toy problems with **soft prompting** and **LoRA** as given in sections: 4.1, A.10.1, 4.2, and A.10.2 we have:

  - The standard model as described in section 3.1 were used.
  - The base model were trained on a context length of 81 input-output pairs, giving the possibility to use **50 soft prompt** pairs plus a context length of 31 for finetuning.
  - **Mean Squared Loss** was the loss function for all runs.
  - A **Learning rate** of **5.0e-05** was used for the base model and **LoRA**. For **soft prompting**, we found that a learning rate of 5e-02 performed better.
  - Pretrained model were trained for **3 million** steps, using **curriculum learning** as described in appendix section A.3.1.
  - Finetuning were done for about **1 million** steps for all methods, though strong performance was often achieved much earlier.
  - The **LoRA** models used rank 4 **LoRA** on the attention matrices, giving **12288 trainable parameters**.
  - The **soft prompting** model with 50 **soft prompt** pairs had **12800 trainable parameters**.
  - The **soft prompting** model with 2 **soft prompt** pairs had **512 trainable parameters**.
  - Evaluations were done on 12800 examples, and 31 input-output pairs.
  - The model pretrained on noisy data (see Appendix A.10.2) was trained for **2 million steps**, reaching convergence.

- For toy problems with alignment as given in sections: 4.3 and 4.4 we have:

  - **Learning rate** of **5.0e-05** were used in all runs.
  - Our pre-trained base model trained exactly as described in **soft prompting** and **LoRA** section as described above.
  - Our large model was trained with an embedding dimension of 256, twelve attention heads, and eight layers. The medium model was trained with an embedding dimension of 128, six attention heads and four layers. Lastly, our small model was trained with an embedding dimension of 64, four attention heads, and two layers.
  - For each graph, we compute the MSE over 1000 polynomials for each context length, and the median is graphed as an aggregate metric.
  - For models that are finetuned to alignment, our model is trained for **150 thousand** steps. For the graph displayed in the paper, we use a medium-sized model, with an embedding dimension of 128, six attention heads, and four layers.

- – We use a clamping threshold of 0.5 for all of our alignment tasks. When aligning via finetuning, we use **hinge loss** with weight 100 for points above the clamping threshold.
- – In order to help with our alignment process, we also chose to use hinge loss to disproportionately penalize points misclassified above our alignment threshold.

### A.3.1. CURRICULUM LEARNING

Training of the pretrained models were done using **Curriculum** training (Wu et al., 2021) by increasing the number of input points to the model in steps. The purpose is to learn easier tasks first, and then increase the difficulty. Gradually increasing the number of points was also done by Garg et al in the seminal work on simple function classes as a toy model for in context learning.

### A.3.2. HARDWARE AND TIME USAGE

The hardware configuration varied between different runs, which were run on several computers. Training the base model for 3 million steps with 81 input-output pairs took about 21 hours on a GeForce RTX 3090. Finetuning with **2 soft-prompt pairs** and **50 soft-prompt pairs** for 1 million steps both took about 6.2 hours. For **LoRA** rank 4, finetuning on 1 million steps took 8.9 hours.

For parts of this work GNU Parallel (Tange, 2018) were used. This research also used the Savio computational cluster resource provided by the Berkeley Research Computing program at the University of California, Berkeley (supported by the UC Berkeley Chancellor, Vice Chancellor for Research, and Chief Information Officer).

### A.4. Baselines

For baselines linear regression in Chebyshev basis is done, both without and with some ridge regularization, which would be the optimal estimator under Gaussian noise. We use a ridge parameter of 0.2 for our experiments. A ridge regression parameter of $\lambda$ is optimal if $\mathcal{N}(0, \lambda)$ Gaussian noise were added.

Using linear regression one would theoretically be able to predict the exact polynomial with $n + 1$ examples for a degree $n$ polynomial. That is why the graphs in figure 2 and 3 are cut off, since the linear regression will be able to get $0 \approx 10^{-\infty}$ error.

### A.5. Bootstrap confidence interval

Bootstrap confidence intervals were used for some of the graphs.

The mean is calculated as:

$$\mu_i = \frac{1}{N} \sum_{j=1}^{N} y_{ij}$$

where $y_{ij}$ is the $j$-th data point of the $i$-th evaluation, and $N$ is the total number of evaluations.

For the bootstrap method, we sample with replacement from the dataset, and calculate the mean of each sample to create a sampling distribution of means. The lower and upper limits of this distribution provide an interval estimate of the true mean. We use the the 5% and 95% percentiles.

**Lower ($L_{\text{boot}}$) an Upper ($U_{\text{boot}}$) Bootstrap Limit**

$$L_{\text{boot},i} = \text{sorted}(\{\mu_i^{(b)}\}_{b=1}^{B})[0.05 \times B], \ \ U_{\text{boot},i} = \text{sorted}(\{\mu_i^{(b)}\}_{b=1}^{B})[0.95 \times B]$$

where $\mu_i^{(b)}$ is the mean of the $i$-th evaluation in the $b$-th bootstrap sample, and $B$ is the total number of bootstrap samples. We use $B = 1000$.

## A.6. Code

Our code for training the base model, and doing **soft prompting** and **LoRA** can be found at `https://github.com/MSNetrom/in-context-poly-playground`. This is a fork of code done by another group, which can be found at `https://github.com/in-context-learning-2024/in-context`. The alignment code has not yet been added, although this is a work in progress. Thanks to (Garg et al., 2023) for providing inspiration for our code on their GitHub: `https://github.com/dtsip/in-context-learning`. Their code is published under the **MIT License**. Also thanks to the **Hugging Face** team for providing code and infrastructure for the GPT2-model (Wolf et al., 2020).

## A.7. Effect of Positional Encoding

Figure 8 shows that the difference in performance between using positional encodings and not using positional encodings seems insignificant. This is not surprising given that the input data is invariant regarding position.

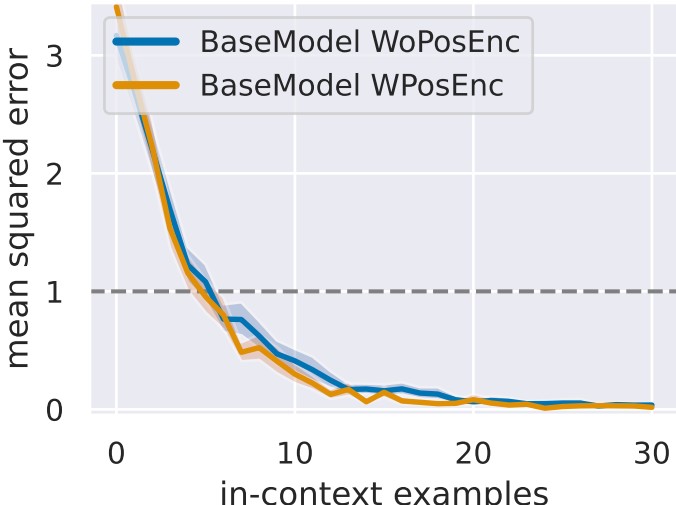

Figure 8: Performance of pretrained models with and without positional encodings on the Chebyshev kernel linear regression task with random degrees between 1 and 11. They are trained as described in section 3.1 and A.3. (**Shaded areas**) represent the 95% bootstrap confidence interval. 1280 samples used.

## A.8. Effect of Learning Rate on Performance for Many Soft Prompts

In Figure 3 in section 4.2, we see that **soft prompting** performs poorly when there is a low number of fixed coefficients, and the distribution of $y$ appears to be fairly independent of $x$. It is discussed if the model might need to learn to ignore the randomly initialized prompts, since the task could be very similar to the original one, leaving little room for learning through prefix-tuning. An alternative explanation is that there are problems with the learning rate. Because of this, a comparison of learning rates are done in Figure 9. Here it is observed that changing the learning rate does have a significant impact, however it does not solve the problem of bad performance. For experiments in this paper, the learning rate of $5e-2$ were chosen based on Figure 9.

## A.9. Linear Scale Prompting Performance Graphs

Figure 10 contains the same graph of pretrained model performance as Figure 2, but with a linear scale. Figure 11 shows the same comparison of **soft prompting** and **LoRA** finetuning methods as Figure 3, but with a linear scale.

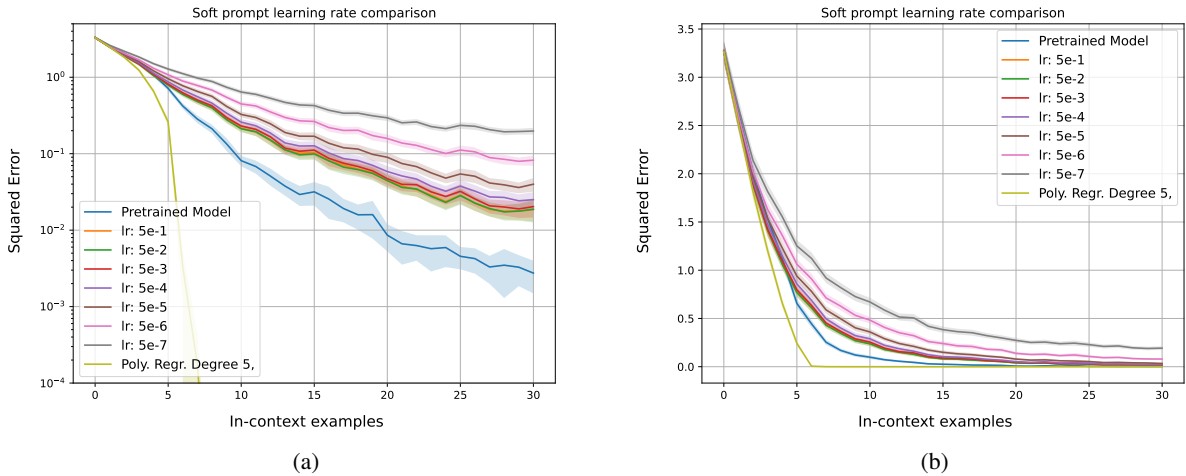

(a)                                                                (b)

Figure 9: Comparison of the effect of different learning rates on **soft prompts**, with **50 soft prompt pairs**. This was done on the task of predicting y-values on normal random linear combinations of Chebyshev polynomials of degree 5, and 0 fixed coefficients. (a) Log-scale used (b) Linear scale used.

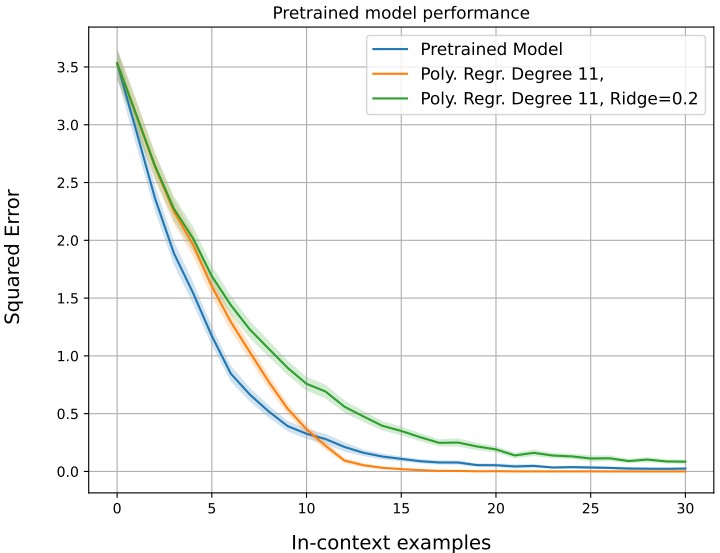

Figure 10: Performance of pretrained model on normal random linear combinations of Chebyshev polynomials of a random degree between 0 and 11. **Polynomial regression** and **ridge-regularized polynomial regression** are used as baselines (See section A.4). For degree 5, 6 examples are theoretically needed to achieve $0 \approx 10^{-\infty}$. The plot is therefore cutoff below. (**Shaded areas**) represent the 95% bootstrap confidence interval (A.5).

### A.10. Extra Finetuning results

#### A.10.1. SOFT PROMPTS VS HARD PROMPTS

To see how the relationship between **soft prompts** and hard prompts transfers to the toy polynomial regression domain (Figure 12), we did some analysis of the learned **soft prompts** for the task described in 3.4. That is polynomial of degree 5, no noise, and 2 fixed linear coefficients. Only 2 pairs of **soft prompts** were used (x, y pairs). Our results match (Bailey et al., 2023), where they observed that in language models, the closest hard prompt (in this case, projection from **soft**

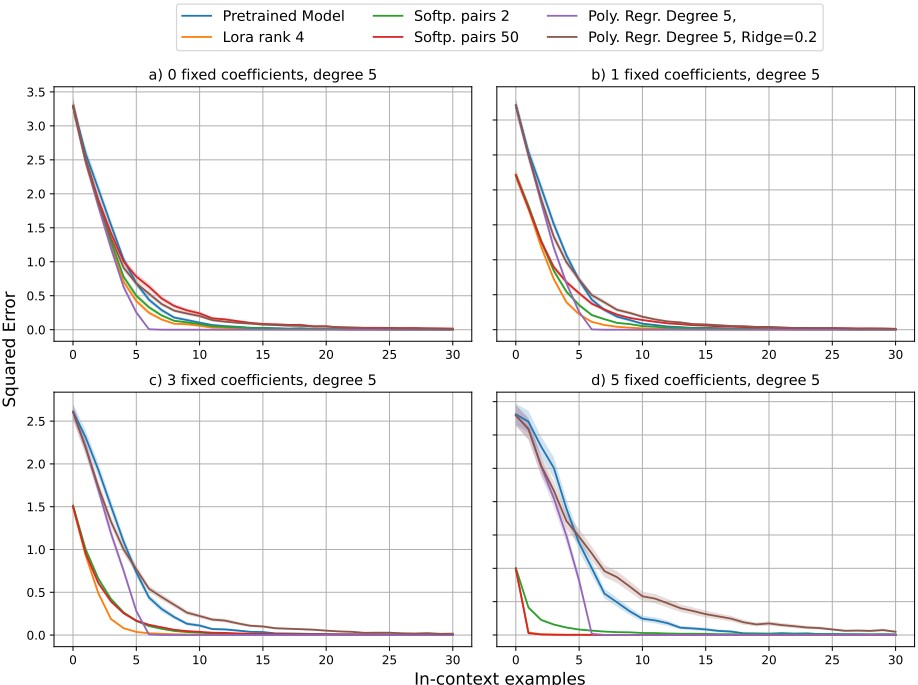

Figure 11: Comparison of performance of finetuning methods on normal random linear combinations of Chebyshev polynomials of degree 5, and with the $n$ first linear combination coefficients fixed. $y$ is more dependent on $x$ with more fixed coefficients. **Polynomial regression**, with and without ridge term, are used as baselines. (a) 0 fixed coefficients, (b) 1 fixed coefficient,(c) 3 fixed coefficients, (d) 5 fixed coefficients. (**Shaded areas**) represent the 95% bootstrap confidence interval (A.5).

**prompt** space to hard prompt space since embedding layer in linear) does not preserve task efficacy at all. We also see high difference magnitudes, where the magnitude of the **soft prompt** is about 40, but the magnitude of the closest hard prompt (in the embedding space) is about 10. Finally, we also see that the prompt does seem a bit more sensitive to rotation compared to scaling, matching (Bailey et al., 2023), as shown in Figure 12. We tried scaling of between 0.5 and 1.2, and rotated between -0.8 and 0.8 radians towards/away from the closest hard prompt. Decreasing magnitude by as much a factor of 2 led to only minimal performance degradation, while increasing magnitude was much worse for performance. This has some agreement with (Bailey et al., 2023), although their language model seemed more robust, as a magnitude reduction of 6 could lead to accuracy improvements. Potentially this difference relates to our much smaller model size and lack of diversity in model training data.

### A.10.2. FINETUNING ON NOISY DATA

We also benchmarked our models on the same Chebyshev polynomials of degree 5 with fixed coefficients as seen in Figure 13, but this time added Gaussian noise sampled from a $\mathcal{N}(0, 0.5)$ distribution, in contrast with noiseless pretraining data. We hypothesized that **soft prompting** might struggle with finetuning on noisy polynomials, since this is a form of task that does not make any clear change in the distribution between x and y. However, it still managed to provide significant improvement over the untrained model. We note that **LoRA** still does perform better though, astonishingly matching the performance of the optimal ridge estimator in the zero fixed coefficients regime, and attaining near zero loss, while the **soft prompt** models never beat the ridge regression estimator, despite additional information about fixed coefficients. This supports the notion that it is harder for **soft prompting** to capture nuanced distributional differences.

### A.11. Extra Alignment Results

In addition to comparing performance across varying context lengths, we additionally test how model size plays a role in the ability to perform this clamping in-context (Figure 14). It seems that once our model crosses a threshold of certain size, it

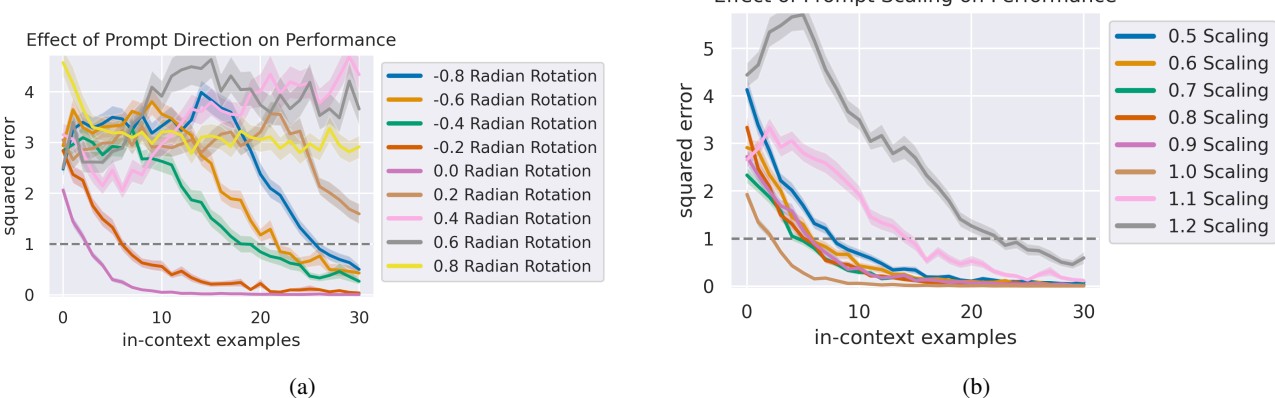

(a)  (b)

Figure 12: Error on Chebyshev task with no noise and 2 fixed coefficients. 2 **soft prompt** pairs are used, and the **soft prompts** are perturbed in terms of scale (12b) and rotated towards the closest hard prompt (12a) by the given amount of radians, negative radians indicating rotation in the other direction. We see increased sensitivity to increasing the magnitude compared to decreasing the magnitude, as well as greater sensitivity to changes in direction. (**Shaded areas**) represent the 95% bootstrap confidence interval (A.5). 1280 samples used.

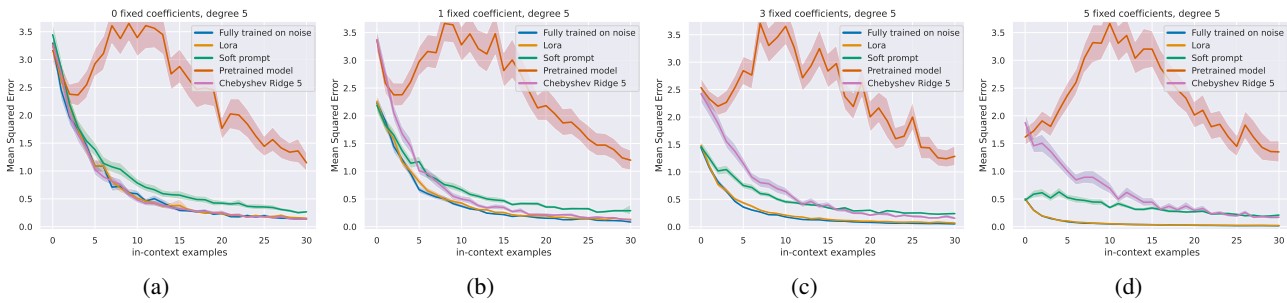

(a)  (b)  (c)  (d)

Figure 13: Performance of **LoRA** and **soft prompting** on data corrupted with $\mathcal{N}(0, 0.5)$ Gaussian noise on combinations of Chebyshev polynomials up to degree 5, and with the $n$ first linear combination coefficients fixed. Note that the loss plotted is to the ground truth, so it is theoretically possible to attain zero loss. **Chebyshev Ridge 5** is Least Squares Regression with ridge regularization (See section A.4). (13a) 0 fixed coefficient (13b) 1 fixed coefficient (13c) 3 fixed coefficients (13d) 5 fixed coefficients (**Shaded areas**) represent the 95% bootstrap confidence interval (A.5). 1280 samples used.

becomes expressive enough to learn better features that are capable of learning from a shifted context.

One question to consider is why is it worth studying clamping of polynomials as a proxy for alignment in the first place? One experiment that helps shape this is to compare the performance of polynomial clamping in-context with other LLM tasks, and see how trends compare (Figure 15). To analyze this, we compare the performance of our polynomial clamping in-context with other common LLM tasks, such as planning, summarization, and translation, all found in previous work, and notice that the trends are similarly monotonically increasing.

In addition to the plot for degree eight provided in the main section of the paper, we include plots for eval-time context clamping for degrees two and four, where we observe similar trends. We additionally include plots of polynomial interpolations, where we visually see the clamped polynomial that our transformer learns via the context. Lastly, we include a plot that validates the foundation of our findings above by confirming that our finetuned-aligned model indeed learns to clamp polynomials above the threshold.

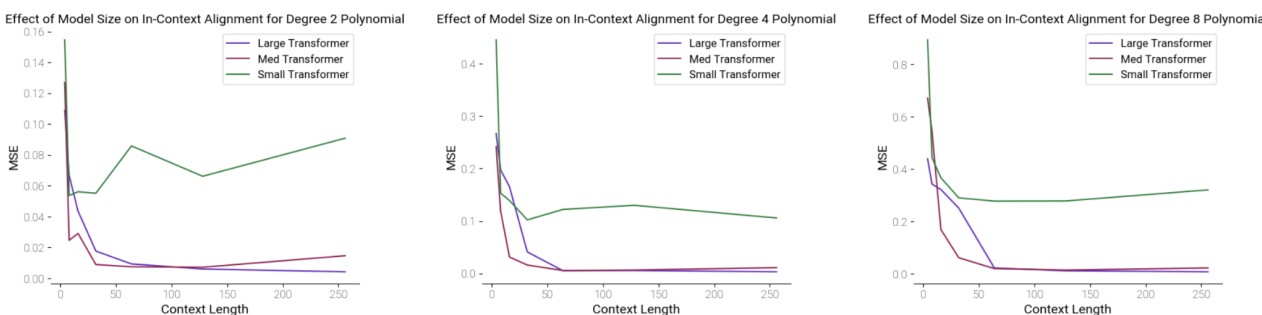

Figure 14: Effect of Model Size on In-Context Alignment. The medium and large transformers perform much better than the small transformer (see appendix section A.3 for model sizes). Each model size decreases in embedding dimension, layers, and heads by a power of two. The results indicate that there is a certain threshold where the model becomes expressive enough to be able to learn alignment behavior from just the context.

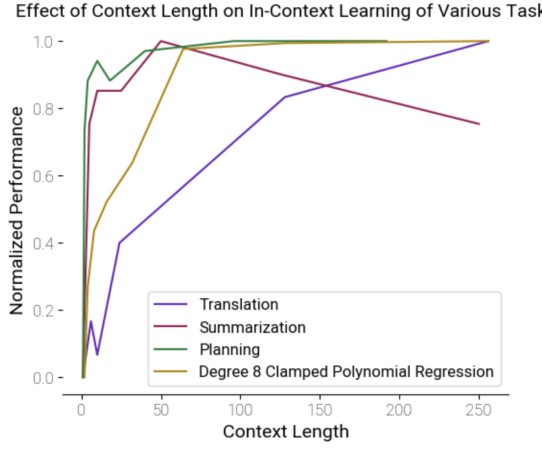

Figure 15: Why Polynomials? Comparing Trends Across Various ICL Tasks
This plot compares performance on translation, summarization, planning tasks from (Agarwal et al., 2024). For the polynomial regression task, we add clamped values to the context as a proxy for alignment to a new task. Trends across very different tasks which attempt to learn a task entirely from the context window (summarization, translation, planning), match that of polynomial regression. To compare the trends across various tasks, the success rate of each task is normalized to its min and max from zero to one.

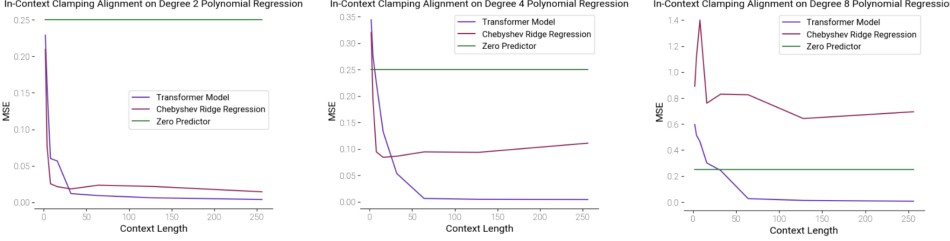

Figure 16: Can Polynomials Be Aligned In-Context?
As the context length increases, the model's performance on points above the threshold increases, whereas the same cannot be said for points below the threshold. Additionally, the negative MSE for points below the threshold (no clamping) is significantly worse than if no clamping is provided in the context window. This hints at the sensitivity to the model to its context in impacting original performance.

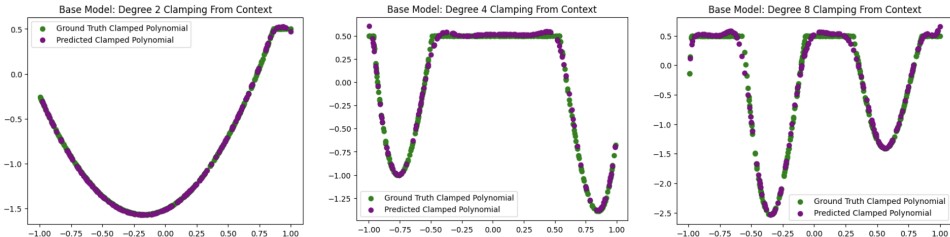

Figure 17: How Well Can We Align Polynomials Given A Clamped Context
Across all degrees, adding in a clamped context leads to good predictions in which values above a threshold of 0.5 are clamped, while the rest of the polynomial shape remains in tact.

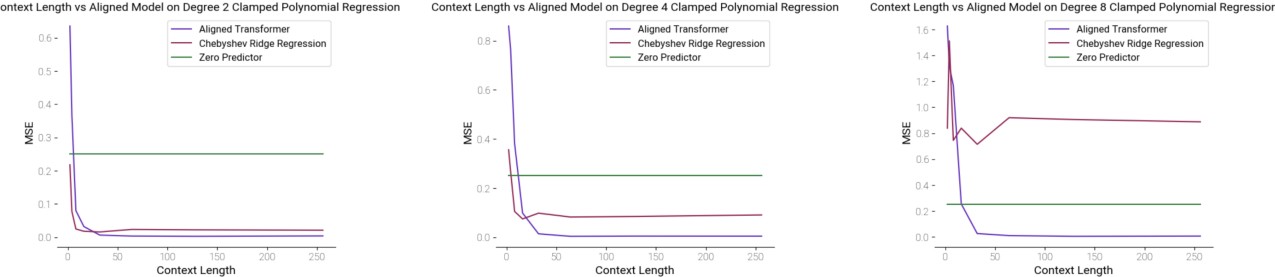

Figure 18: Clamped Alignment of Base Polynomial Regression Model
The transformer model is able to perform the task of clamping values in-context after being additionally finetuned. It predictably outperforms the Chebyshev Ridge and zero predictor baselines, and this performance difference is especially visible at higher degrees.

