# OpenReview forum: "Polynomial Regression as a Task for Understanding In-context Learning Through Finetuning and Alignment"
_ICML.cc/2024/Workshop/ICL — ICML 2024 Workshop ICL Poster_

### Official Review · Reviewer_Gbz8 · 2024-06-07
**An interesting and creative approach to studying fine-tuning and alignment in Transformers using a very simple task**

**Rating:** 2
**Fit:** 3
**Confidence:** 2

**Workshop Review:**

This publication presents a creative analysis of fine-tuning and alignment in Transformers by experimenting with in-context univariate polynomial regression. While extremely simple, this setup gives the authors some flexibility in empirical investigations of LoRA and soft prompt model tuning and in the analysis of alignment and model jailbreaking. The approach appears to be novel and creative and can be of interest to the community. Quantitative and qualitative empirical observations may not be directly transferable to practical models, but show some signs of intriguing similarity. Overall, the paper appears to be well-grounded and correct, however it is not clearly written and some sections (like Section 3.4) can be open to interpretation. There are additional minor concerns outlined below.

**Reason For Not Giving Higher Score:**

The paper is not entirely clearly written and some sections (like Section 3.4) are a little confusing and allow multiple interpretations. I am also confused with the discussion of experimental results. For example, in Section 4.1 the authors state that "the model performs
better than two relevant baselines", however Figure 2 seems to show that Chebyshev 11 actually outperforms a pretrained model when the number of in-context samples exceed ~12. I am not sure how to interpret this discrepancy.

In Section 4.2 the authors highlight a poor performance of a model with 50 soft prompt tokens. The authors attribute this to the fact that these 50 tokens "effectively need to be ignored", but it is not clear if this is such a complex task. If the soft prompt is unconstrainted, it is conceivable that it would converge to nearly-identical vectors with a large negative $\vec{k}\cdot \vec{q}$ product at every layer thus having virtually no impact on the attention matrix for the actual sequence tokens. From my experience, when training soft prompts, one has to be careful with the learning rate choice (the optimal learning rate is typically sensitive to what exactly is being tuned) and the choice of other parameters. Perhaps the authors could include additional evidence in Appendix that would support the result and alleviate such suspicions.

Another minor issue relates to Figures 2 and 3. They depict the evolution of several positive quantities that converge to 0 and it becomes difficult to see the difference between them. In my opinion, both plots should use logarithmic $y$ scale.

**Reason For Not Giving Lower Score:**

Overall, the paper proposes a very simple, but meaningful and creative setup for studying fine-tuning and alignment in transformers. It provides sufficiently well-supported empirical results that appear to agree with similar studies in large practical models (including LLMs). While it is difficult to draw immediate conclusions from these observations they can perhaps be expanded upon and be analyzed theoretically in the follow-up work.

---

### Official Review · Reviewer_fZtu · 2024-06-08
**Thorough analysis on an elegant toy problem that might have interesting qualitative parallels with empirical observations in LLMs**

**Rating:** 2
**Fit:** 3
**Confidence:** 2

**Workshop Review:**

This paper considers the problem of polynomial regression in the Chebyshev basis. After training a transformer model with a suitable curriculum to induce ICL behavior, the paper probes model behavior under variations of prompting and fine-tuning that mimic analyses in LLMs, including a mock version of alignment.

The chosen toy problem is elegant and the analysis is thorough. The paper is clearly written, and elaborates on the experimental details which is crucial given the empirical nature of the work. The work should therefore be of interest to the community.

**Reason For Not Giving Higher Score:**

Given the empirical focus of the analysis (rather than any mechanistic understanding of the phenomena) it is unclear how these insights might transfer to more general problems of interest.

**Reason For Not Giving Lower Score:**

The chosen toy problem is elegant and the analysis is thorough. The paper is clearly written, and elaborates on the experimental details which is crucial given the empirical nature of the work.

---

### Author Response · Authors · 2024-07-26
**Update on camera ready version**

We would like to thank the reviewers for their informative and insightful feedback. This helped us improve our paper.
We have made several improvements to the paper:
- We tested several learning rates for soft-prompting, and added the performance to the appendix. This did in fact have an impact on the performance, although not significant enough of an impact to change our interpretation of the results. It still seems plausible that for the tasks that are similar to the pretraining task, the model must learn 50 soft prompts which can be ignored, and this seems like it could be difficult.  We also show that fixing 5 coefficients makes all polynomials have 5 shared fixed points. This is a task where soft-prompting performs well, and this supports that soft-prompting might benefit from a task with a more predictable distribution.
- We have rephrased section 4.1 to fix this discrepancy: "the model performs better than two relevant baselines.”
- We changed to log-scale for figures where the error converges to 0.
- We added a link to our codebase.
- We improved the analytical derivation of our distribution of x-y values for the task used in the pretrained model included in the appendix.
- We made parts of the paper more clear, like section 3.4.
- We improved alignment figure 4b, with more details in the legend.

---

### Meta-Review · Area_Chair_dmVM · 2024-06-12

**Recommendation:** 2

**Metareview:**

The paper "Polynomial Regression as a Task for Understanding In-context Learning Through Finetuning and Alignment" introduces univariate polynomial regression as a novel toy problem for exploring phenomena of in-context learning (ICL) in large language models (LLMs).

This paper presents an interesting and innovative approach to studying fine-tuning and alignment in Transformers using a simple yet insightful task. Both reviewers appreciated the elegance and creativity of the toy problem and recognized its potential value to the community. The empirical results, while not immediately transferable to practical models, provide intriguing parallels with observations in LLMs.

The primary concerns raised were about the clarity of writing and the presentation of experimental results. The authors are encouraged to address these concerns, particularly the clarity of Sections 3.4 and 4.1 and the visualization of data in Figures 2 and 3. Providing additional evidence in the Appendix to support their findings on soft prompting would also strengthen the paper.

Based on the reviewers' comments and the overall quality of the work, this paper should be accepted as a poster presentation.

---

### Decision · Program_Chairs · 2024-06-17

Accept (Poster)